# The Ambivalent Role of miRNA-21 in Trauma and Acute Organ Injury

**DOI:** 10.3390/ijms252011282

**Published:** 2024-10-20

**Authors:** Aileen Ritter, Jiaoyan Han, Santiago Bianconi, Dirk Henrich, Ingo Marzi, Liudmila Leppik, Birte Weber

**Affiliations:** Department of Trauma, Hand and Reconstructive Surgery, University Hospital Frankfurt, Goethe University, 60486 Frankfurt am Main, Germany; j.han@med.uni-frankfurt.de (J.H.); bianconi@med.uni-frankfurt.de (S.B.); d.henrich@trauma.uni-frankfurt.de (D.H.); marzi@trauma.uni-frankfurt.de (I.M.); lleppik@yahoo.com (L.L.); bi.weber@med.uni-frankfurt.de (B.W.)

**Keywords:** miR-21, polytrauma, TBI, trauma, miRNA, cardiac damage, lung injury, spinal cord injury, nerve trauma, osteoporosis, fractures

## Abstract

Since their initial recognition, miRNAs have been the subject of rising scientific interest. Especially in recent years, miRNAs have been recognized to play an important role in the mediation of various diseases, and further, their potential as biomarkers was recognized. Rising attention has also been given to miRNA-21, which has proven to play an ambivalent role as a biomarker. Responding to the demand for biomarkers in the trauma field, the present review summarizes the contrary roles of miRNA-21 in acute organ damage after trauma with a specific focus on the role of miRNA-21 in traumatic brain injury, spinal cord injury, cardiac damage, lung injury, and bone injury. This review is based on a PubMed literature search including the terms “miRNA-21” and “trauma”, “miRNA-21” and “severe injury”, and “miRNA-21” and “acute lung respiratory distress syndrome”. The present summary makes it clear that miRNA-21 has both beneficial and detrimental effects in various acute organ injuries, which precludes its utility as a biomarker but makes it intriguing for mechanistic investigations in the trauma field.

## 1. Introduction

In 1993, the working group of Lee found a gene that was not a protein, but a small RNA (ribonucleic acid) playing a role in larval development [1]. In 2001, these types of RNAs were called microRNAs (miRNAs) and were recognized as molecules functioning in the regulation of translation. These microRNAs were found to exhibit a wide diversity of functions, for example, in differentiating tissue expression [2,3]. These single-stranded RNAs are non-coding, at a maximum of 22 nucleotides long, and are encoded by short repeats in the genome [4].

Since then, miRNAs have been recognized to play an important role in the mediation of various diseases and are considered potential biomarkers. In particular, in the field of cancer diagnosis and treatment, miRNAs have been employed as diagnostic and even prognostic biomarkers. miRNAs were described to either stimulate tumor growth and differentiation or inhibit tumor growth and regulate metastasis development [5,6]. Furthermore, they are important actors in cell–cell-communication, because they can also be transported in exosomes [7].

One of the earliest miRNAs to be discovered, miR-21, is found in the intron of the transmembrane protein 49 (TMEM49)/vacuole membrane protein 1 (VMP1) locus located in human chromosome 17 [8]. Compared to other miRNAs, the regulation of miR-21 is more complex and is influenced by cytokines and hypoxia [9]. The transcribed product, pri-miRNA (3433 nucleotides), proceeds to form pre-miR-21 (72 nt), which gives rise to both miR-21-3p and miR-21-5p. Phosphatase and Tensin Homolog (PTEN), Tropomyosin 1 (TPM1), and Programmed Cell Death 4 (PDCD4) are the three main targets out of more than 3000 target genes of miR-21. Many of these genes have a variety of functions in many disorders, including cancer [10,11,12]. Although miR-21 is one of the most studied miRNAs in disease, its function in traumatic injury is still unclear. One of the reasons for this could be that most research does not differentiate between miR-21-5p (guide strand) and miR-21-3p (passenger strand), two miRNAs which could function differently and have different target genes.

Therefore, the main goal of the present review is to summarize the roles of miR-21-5p and miR-21-3p in trauma and acute organ injury Figure 1. The literature search for this review is based on a PubMed search with the terms “miRNA-21” and “trauma”, “miRNA-21” and “severe injury”, and “miRNA-21” and “acute lung respiratory distress syndrome”. The literature search was conducted in May of 2024; therefore, studies up to this time point were included in this review. 

## 2. miR

### 2.1. MiR-21 in Traumatic Brain Injury

Worldwide, severe traumatic brain injury (TBI) and polytrauma remain major causes of death, especially for younger patients (age < 45 years) [13]. Over the last few decades, TBI-related mortality has been a leading cause of death in trauma patients [14,15].

Our literature search showed that TBI belongs to the most analyzed acute organ injury with regard to miR-21.

### 2.2. In Vitro Studies

The positive role of miR-21 in TBI was shown in several in vitro studies mimicking brain injuries on brain endothelial cells. Brain microvascular endothelial cells (BMVECs) are the major component of the blood–brain barrier and are widely used to examine molecular processes after TBI. Upregulation of miR-21 in response to scratch injury was shown to alleviate damage to the endothelial barrier and the loss of tight junction proteins [16,17]. The impacts of miR-21-5p on the activities of NF-kB (nuclear factor “kappa-light-chain-enhancer” of activated B cells), Akt (protein kinase B), and Ang-1/Tie-2 (angiopoietin-1/tyrosine kinase receptor) signaling [16], as well as the activation of the Ang-1/Tie-2 axis [17], were described. In another in vitro TBI scratch injury model using PC12 cells (pheochromocytoma cell line), the upregulation of miR-21, p-PI3K (phosphoinositide 3-kinase), p-Akt, and p-GSK-3β (glycogen synthase 3β) was shown [18].

Therapeutic transfection of miR-21 in apoptotic neurons (TBI-induced) was found to reduce the number of TUNEL-positive neurons, decrease the expression level of PTEN, and increase phosphorylation of Akt [19]. In transfected cells, miR-21 agomirs enhance the expression of anti-apoptotic operating Bcl-2 (B-cell leukemia/lymphoma 2 protein) and suppress the expression of apoptotic operating caspase-3, caspase-9, and Bax [19].

In contrast, in a coculture model of PC12 neurons and BV2 microglia cells, the negative role of exosomal miR-21-5p was shown. Thus, PC12-derived exosomes containing miR-21-5p were found to aggravate the release of neuroinflammation factors, inhibit neurite outgrowth, increase the accumulation of P-tau, and promote the apoptosis of PC12 cells [20].

The roles of miR-21 have also been investigated in ischemic brain injury. Lopez et al. showed that exposure of cortical primary neurons and astrocytes to hypoxia and glucose deprivation induced a two-fold increase in miR-21 in these cells [21].

In another in vitro model, in which an injured brain microenvironment was mimicked by treatment of HT-22 neurons with brain extracts harvested from a TBI mouse, a significant increase in miR-21-5p in exosomes from treated cells was found. It was demonstrated that these miR-21-5p-enriched neuronal exosomes reduced trauma-induced, autophagy-mediated nerve damage by inhibiting neuronal autophagy activity via the targeting of Rab11a [22].

### 2.3. In Vivo Studies

According to in situ localization studies, miR-21 is widely expressed in the normal brain and is upregulated in the cortex and hippocampus, particularly the dentate gyrus and CA3 cell layer, following TBI [23]. Sandhir et al. described that miR-21 expression profile in mice correlated with age, as basal miR-21 expression was higher in the aged brain than in the adult brain. In the adult brain, miR-21 expression increased in response to injury, with the maximum increase 24 h after injury, followed by a gradual decrease and return to baseline 7 days post-injury. In contrast, in aged mice, miR-21 showed no injury response, and expression of miR-21 target genes (*PTEN, PDCD4, RECK, TIMP3*) was upregulated at all post-injury time points, with a maximal increase at 24 h post-injury [24].

The kinetics of miR-21 expression after TBI were also investigated by Redell et al. in a controlled cortical impact model in rats. After TBI, miR-21 expression was significantly upregulated in the hippocampus, reached expression maximum at 3 days post-injury, and returned to near pre-trauma levels 15 days later [23]. At the same time, 10% penetrating unilateral frontal ballistic-like brain injury induced increases in miR-21 4 h and 1 day later. This elevation was sustained until 7 days after trauma [25]. In adult male rats, a unilateral controlled cortical impact over the sensorimotor cortex was shown to induce an acute increase in superoxide dismutase 2 mRNA, miR-21, and miR-155 expressions, which have been previously demonstrated to disrupt mitochondrial homeostasis [26]. A microarray study in a controlled cortical impact model in C57Bl/6 mice further proved a significant increase in miR-21, miR-144, miR-184, miR-451, and miR-2137 and a decrease in miR-107, miR-137, miR-190, and miR-541 at 1, 6, and 12 h after TBI [27]. Moreover, another study found a global upregulation of miR-21 expression in the rat cortex after TBI throughout all analyzed time points post-injury (6 h, 24 h, 48 h, and 72 h) [28].

Interestingly, it was found that the global upregulation of miR-21, miR-92a, and miR-874 and downregulation of miR-138, let-7c, and miR-124 in pre-trained mice were linked to a significant decrease in the mortality rate and improved recovery in TBI mice [29]. The same working group found altered miRNA expressions in the hippocampus of mice following TBI (days 15 post running wheel exercise). The results showed that spontaneous exercise recovered the hippocampus-related cognitive deficits induced by TBI, and altered hippocampal expressions of miR-21 and miR-34a associated with the recovery process [30]. Ge et al. showed that upregulated miR-21 conferred a better neurological outcome and alleviated TBI-induced secondary blood–brain barrier (BBB) damage and loss of tight junction proteins [17]. The anti-apoptotic therapeutic effect of exosomal miR-21-5p was described in a rat model of subarachnoid hemorrhage (SAH). The transfer of mesenchymal cell (MSC)-derived EVs to an SAH reduced the apoptosis of neurons during early brain injury and induced further increases in miR-21 expression in the prefrontal cortex and hippocampus [31].

Ge et al. found that the intracerebroventricular infusion of miR-21 agomirs inhibited apoptosis and promoted angiogenesis in a fluid percussion injury rat model. In line with the above-mentioned results, the upregulation of miR-21 in the brain conferred a better neurological outcome after TBI by improving long-term neurological function, alleviating brain edema, and decreasing lesion volume [32].

Additionally, the protective role of miR-21-3p and the alleviation of BBB leakage were shown in a TBI mouse model. In the treatment group, Evans Blue extravasation was reduced and expression of tight junction proteins was promoted, thus contributing to the improved neurological outcome of controlled cortical impact (CCI) mice [33]. MiR-21-3p antagomirs suppressed cell death by preventing apoptosis and controlling the inflammatory response through the inhibition of NF-κB activity in a cortical impact injury in mice [33].

Furthermore, miRNA-21 was shown to have positive effects after stroke: in mice with transient middle cerebral artery occlusion (C57BL/6 mice), intravenous or local application of miR-21 mimics decreased post-ischemic levels of several pro-apoptotic and pro-inflammatory RNAs, which might be responsible for neuroprotection [34].

### 2.4. Patient Studies

In patients, Pinchi et al. found that expression levels of miR-21-5p, miR-92, and miR-16 in brain tissue had a high predictive power in discriminating brain trauma cases from controls in patients with a fatal outcome [35]. Furthermore, Di Pietro et al. suggested that serum levels of miR-21 and miR-335 could help in the diagnosis of severe TBI [28,36]. miR-21-5p was found to be the most abundant miRNA in the cerebrospinal fluid (CSF) of severely injured TBI patients [37]. In a prospective case–control study, miR-21-5p and miR-221 were found to be increased in cerebrospinal fluid in subarachnoid hemorrhage (SAH) patients with delayed cerebral ischemia [38].

In summary, miR-21 may have a positive effect on the course of recovery following TBI by anti-apoptotic effects via the increased production of anti-apoptotic Bcl-2 and effects on the inflammatory response [19,32,33] (see Table 1 for a summary). miR-21 is significantly upregulated in the serum of TBI patients and was associated with better neurological outcomes after TBI in a rat model [17,36]. These findings suggest its potential as a TBI outcome biomarker. However, one has to be careful regarding the previously mentioned different variants of miR-21. While miR-21-5p seems to have mostly positive effects after TBI, e.g., alleviating damage to the endothelial barrier [16,17], there are some clues that miR-21-3p seems to be detrimental, as the injection of miR-21-3p antagomirs alleviated the loss of the BBB in mice [33]. The summarized role of miR-21 in the context of TBI is demonstrated in Table 1.

## 3. miRNA-21 in Spinal Cord Injury

The incidence of SCI worldwide ranges from 10 to 83 cases per million persons per year [39]. SCI is more common in younger adults and is frequently triggered by violence or car accidents [40,41]. In general, miR-21 was shown to be increased after SCI starting from day 1, up to 28 days after injury. Overexpression of miR-21 during SCI is associated with many protective effects like the inhibition of neuroinflammation, the improvement of blood–spinal cord barrier function, the regulation of angiogenesis, and the controlling of glial scar formation (reviewed in [42]/summarized in Table 2).

### 3.1. In Vitro Studies

There are only a few in vitro studies evaluating the roles of miR-21 in SCI. miR-21 exhibited anti-apoptotic effects in SCI by downregulating PTEN [42]. The therapeutic effect of this miRNA in SCI was shown in in vitro experiments with the use of short hairpin RNA-loaded exosomes [43]. Furthermore, miR-21-5p was shown to recover remyelination in inflammation-treated neurons in in vitro coculture experiments with neurons and neuronal stem cells. Neurons which were treated with LPS first and then with miR-21 inhibitors/antagomirs produced EVs with upregulated miR-21. These EVs function through SMAD 7 (small worm phenotype/mothers against decapentaplegic of genes in *Drosophila*)-mediated activation of the TGF-β (transforming growth factor)/SMAD 2 signaling pathway [44].

### 3.2. In Vivo Studies

The beneficial effects of miR-21 were observed in various in vivo SCI studies. In a spinal contusion rat model, miR-21 was found to be significantly upregulated at day 1 and day 3 after trauma. After knockdown of miR-21, recovery in motor function was delayed, and the size of the lesion and apoptosis were increased [45]. Intravenous injection of PC12- and MSC-derived EVs enriched with miR-21/mi-R19b was shown to suppress neuron apoptosis by downregulating PTEN expression [46]. Additionally, EVs from miR-21-transfected MSCs improved the functional recovery of SCI rats and suppressed cell death in vivo and in vitro via the miR-21/PTEN/PDCD4 signaling pathway. miR-21 was shown to inhibit the expression of PTEN/PDCD4 and suppress neuron cell death in this study [47].

In an SCI rat model, He et al. induced miR-21 expression in spinal cord neurons using an miR-21 lentiviral vector. This improved neuronal survival and promoted functional recovery after injury. Moreover, the overexpression resulted in reduced cellular apoptosis through a decrease in PDCD4 protein and caspase-3 activity [48]. Overexpression of miR-21 in astrocytes attenuated the hypertrophic response to SCI in transgenic mice [49].

Studies focusing on different therapeutic options for SCI further showed the importance of miR-21. In a modified Allen’s weight drop model, the use of miR-21 agomirs modulated the secretion, proliferation, and apoptosis of astrocytes and promoted recovery after SCI both in vivo and in vitro. These effects were likely mediated by TGF-β targeting the PI3K Akt/mTOR (mammalian target of rapamycin) pathway [50]. For treatment options, tetramethylpyrazine (TMP) was shown to be a promising tool. TMP treatment for 3 days after SCI significantly improved functional recovery, decreased lesion size, and increased tissue sparing and miR-21 expression, while TMP decreased the expression of FasL (Fas ligand), PDCD4, and PTEN [51]. In a rodent SCI model, Wang et al. pre-conditioned animals with sevoflurane and showed that this increased the expression of miR-21-5p and decreased PPP1R3B and MAP2K3 proteins. In control animals, SCI induced significant underexpression of miR-21-5p and upregulation of PPP1R3B (protein phosphatase 1 regulatory subunit 3B) and MAP2K3 proteins [52]. Other experiments investigated Gypenoside XVII (GP-17) as a therapeutic option in an SCI mouse model. The results showed that SCI downregulated miR-21, while GP-17 treatment upregulated it. The inhibition of miR-21 eliminated the protective effects of GP-17 on SCI-induced neuronal apoptosis and the inflammatory response. The findings of this study suggest that GP-17 plays a protective role in SCI because GP-17 regulates the miR-21/PTEN/AKT/mTOR pathway [53]. A reduction in phosphatase and PTEN has been shown to be involved in axonal regeneration and synaptic plasticity in a rat model with cervical hemisection after a single bolus of docoshexaenoic acid (DHA). The authors showed that DHA significantly upregulated miR-21 and downregulated PTEN in corticospinal neurons. Downregulation of PTEN and upregulation of phosphorylated AKT by DHA were also seen in primary cortical neuron cultures. This was accompanied by increased neurite outgrowth [54]. The protective functioning of miR-21-5p can be boosted with a methylprednisolone injection as Abdallah et al. suggested. In a rat model of spinal trauma, they showed that methylprednisolone increased miR-21 expression in the early period of trauma [52].

The effect of miR-21-containing EVs on recovery after SCI was also tested in a rat SCI model. MSC-derived EVs were injected via the tail vein and were found to be directed to the site of spinal injury and mainly incorporated into neurons within the lesioned site of the spinal cord. EVs locally attenuated the lesion size and apoptosis and led to a significant improvement in functional recovery. miR-21-5p was one of the highly expressed miRNAs in these EVs, and its inhibition reversed the beneficial effects of MSC-EVs on motor function and apoptosis [55]. In contrast, Wang et al. found that miR-21-5p promoted the pro-fibrogenic activity of TGF-β1 in spinal fibroblasts and scar formation in an SCI mouse model, whereas the knockdown of miR-21-5p attenuated this activity [56]. Ning et al. showed that the downregulation of miR-21 in SCI rats led to better movement and coordination after injury. Inhibition of miR-21 decreased the protein levels of iNOS (inducible nitric oxide synthase) and TNF-α (tumor necrosis factor alpha) and the RNA levels of IL-6R (interleukin 6 receptor), JAK (janus-activated kinase), and STAT3 (signal transducer and activator of transcription 3) [57].

### 3.3. Patient Studies

Up until today, no research focused on miR-21 expression has been conducted on patients with SCI.

In most of the SCI studies, miR-21 was associated with protective effects after injury, as it inhibits neuroinflammation, improves blood–spinal cord barrier function, regulates angiogenesis, and controls glial scar formation. Several SCI-therapeutic tools (Gypenoside XVII, MSC-EVs, agomirs, or methylprednisolone), which function through the miR-21/PTEN/AKT/mTOR pathway, have been successfully tested in vivo (Table 2). At the same time, some studies suggest the fibrogenic activity of miR-21 and the promotion of scar formation in SCI.

This discrepancy in results could be partially explained by the studies’ failure to differentiate between miR-21-5p and miR-21-3p. Further precise research is required.

**Table 2 ijms-25-11282-t002:** Spinal cord injury.

miR-21 in SCI	Increase in miR-21 1 day after SCI and up to 28 days after injury. Overexpression of miR-21: Inhibited neuroinflammation. Improved blood–spinal cord barrier function. Regulated angiogenesis. Controlled glial scar formation. Downregulated PTEN, Spry2, PDCD4. Suppressed neuron cell death. DHA significantly upregulated miR-21 and downregulated PTEN in corticospinal neurons.	[42,43,46,47,54]
Contusion SCI in rats	Upregulation of miR-21. Knockdown of miR-21: Attenuated recovery. Increased lesion size. Decreased tissue sparing. Increased apoptosis.	[45,46]
Mouse model of SCI, treatment (neuroprotection) of Gypenoside XVII (GP-17)	Downregulation of miR-21 expression following SCI. Upregulation of miR-21 following administration of GP-17. Inhibition of miR-21 eliminated protective effects of GP-17.	[53]
SCI rat model, injection of MSC EVs via tail vein	MSC-EVs significantly improved functional recovery. Inhibition of miR-21-5p in MSC-EVs reversed beneficial effects of MSC-EVs.	[49]
Mouse model of SCI In vitro spinal fibroblast culture	miR-21-5p knockdown in a mouse model significantly improved motor functional recovery after SCI. miR-21-5p promoted pro-fibrogenic activity of TGF-β1 in spinal fibroblasts.	[56]
Administration of miR-21 inhibitor in activated microglia cells of rat	Inhibition of miR-21 decreased protein levels of iNOS and TNF-α.	[57]

## 4. Peripheral Nerve Injury/Nerve Trauma

Peripheral nerve injury is a common disorder in clinical practice that can arise from trauma or resulting surgery following trauma [58]. The impairment in sensory function and muscle force reduction are presented in approximately 60% and 47.3% of patients with traumatic lower extremity and lumbosacral peripheral nerve injuries [59]. Despite our advances in understanding the mechanisms underlying injury and regeneration, functional recovery remains unsatisfactory in many patients. This significantly lowers the quality of life for these patients and places a significant financial burden on health care systems [60,61]. Even though studies on peripheral nerve injuries highlighted the importance of miRNAs in the field, the precise role of miR-21 still needs to be investigated. The recent literature has produced contradictory results (Table 3).

### 4.1. In Vitro Studies

In cultured dorsal root ganglia (DRG) neurons, IL-6 was found to upregulate the expression of miR-21 and miR-222, leading to the inhibition of apoptosis and enhanced viability in these cells [62]. Simeoli et al. also investigated DRG from sensory neurons in vitro and suggested that capsaicin activation of TRPV1 (transient receptor potential cation channel subfamily V member 1) receptors induces the release of exosomes enriched with miR-21-5p. After being phagocytosed, these exosomes increased miR-21-5p expression and promoted pro-inflammatory differentiation of macrophages [63].

### 4.2. In Vivo Studies

Overall, the aberrant expression of miRNAs was described in spinal nerve ligation (SNL), dorsal root transection (DRT), and ventral root transection (VRT) rat models, and two miRNAs (miR-21-3p and miR-31) were found to be significantly upregulated [64]. Similar results were published by Yu et al. who analyzed the expression of miRNAs in the DRG at 1, 4, 7, and 14 days after sciatic nerve resection in rats and identified upregulated levels of miR-21 and miR-221 at all time points [65]. Tissue inhibitor of metalloproteinase 3 (TIMP3) was suggested as a possible target of miR-21, mediating anti-apoptotic effects, although it was not clarified whether miR-21-3p or miR-21-5p was analyzed [62].

It has been proposed that miR-21 plays a role in the promotion of Schwann cell (SC) proliferation and axon regeneration after nerve damage. In a rat peripheral nerve injury model, Ning et al. found that animals treated with miR-21 have a significantly higher proliferation index, lower apoptosis rate, longer axons, lower levels of caspases 3 and 9, and mRNA and protein expression of TGFß1, TIMP3, and EPHA4 (ephrin type-A receptor 4). The authors suggest that miRNA-21 plays a significant role in the promotion of SC proliferation and axon regeneration [66] because miRNA-21 controls the target genes of TGF-βI, TIMP3, and EPHA4 [66].

The role of miR-21 in the promotion of axon regeneration was shown in a model of L4 and L5 DRG sciatic nerve transection. Seven days post-axotomy, miR-21 was found to be upregulated 7-fold in the DRG. The authors suggested that miR-21 is an axotomy-induced miRNA that enhanced axon growth by downregulating the Sprouty 2 protein [67]. These outcomes are supported by a recent study by Ikuma et al. investigating the extracellular release of miRNAs from DRG neurons in a rat model of neuropathic pain induced by chronic constriction injury of the sciatic nerve. The release of six miRNAs (let-7d, miR-21, miR-142-3p, miR-146b, miR-203-3p, and miR-221) from primary cultured DRG neurons prepared from rats 7 days after nerve injury was elevated [68].

Moving to another important topic, Sakai et al. examined the effects of neuropathic pain on miR-21-expression. Neuropathic pain was induced in rats through the ligation of the left fifth lumbar spinal nerve. After the injury, miR-21 expression in the injured DRG neurons, but not in the neighboring uninjured DRG neurons, was persistently upregulated following the pain development. The miR-21 expression in the DRG was increased by intrathecal application of interleukin-1β [69].

### 4.3. Patient Studies

Until now, no studies have examined the role of miR-21 in patients with peripheral nerve injury.

All in all, miR-21 seems to play an important role in peripheral nerve injury and the development of neuropathic pain. Higher levels of miR-21 (and especially miR-21-3p [64]) were detected after nerve injury, with interleukins being a possible stimulus for this upregulation. There is currently a deficiency in differentiated studies of miR-21 in nerve damage, particularly in patients.

**Table 3 ijms-25-11282-t003:** Nerve trauma.

Rat dorsal root ganglion (DRG) neurons	IL-6 induces upregulation of miR-21. Inhibition of neuronal apoptosis of miR-21 through suppression of TIMP3.	[62]
Spinal nerve ligation, dorsal root transection, ventral root transection in rats	Upregulation of miR-21 in injured DRG. miR-21 promotes Schwann cell proliferation. miR-21 enhances axon growth.	[64,65,67,68,69]

## 5. Bone Injuries

Almost 60% of polytraumatized patients admitted into the emergency department have at least one fracture of a long bone [70]. Furthermore, it is commonly recognized that as society ages, there is a corresponding rise in the occurrence of osteoporotic fractures, with osteoporosis accounting for about 9.9 million fractures yearly [71]. The number of studies of traumatic and osteoporotic fractures is expanding, and with this, the understanding of the function of miRNAs.

### 5.1. In Vitro Studies

Studies evaluating the role of miR-21 in bone injury and osteoporosis in vitro are scarce. Zhao et al. pointed out that the reversion-inducing cysteine-rich protein with Kazal motifs (RECK) gene could be a target of miR-21 in osteoporosis [72], whereas Yang et al. showed TNFα to have a suppressive effect on miR-21 expression [73]. There is also literature about miRNA 21 in heterotopic ossification. Subramaniam summarized the role of miR-21 in osteogenesis with its different pathways and effects on the expression of osteogenic factors like RUNX2 (runt-related transcription factor 2) and ALP (alkaline phosphatase) [74].

Furthermore, Wang et al. found that in ASCs (adipose-derived stromal stem cells) and L6 cells, overexpression of miR-21-5p led to the upregulation of BMP-4 (bone morphogenetic protein 4). These findings indicate the influence of miR-21-5p on the osteogenic differentiation of ASCs [75]. Another protein that seems to be connected with miR-21 is bone morphogenetic protein 9 (BMP9). As Song et al. pointed out, miR-21 was upregulated in murine multilineage cells (MMCs) during osteogenesis induced by BMP-9. miR-21 even suppressed Smad7 (SMAD family member 7) and therefore promoted osteogenic differentiation [76].

### 5.2. In Vivo Studies

To assess miR-21 function in fractures, Sun et al. performed a local injection of rat bone marrow MSCs overexpressing miR-21 in rats with closed femur fracture and internal fixation. The upregulation of miR-21-5p increased the osteogenic differentiation of MSCs via the enhanced expression of osteopontin, through alkaline phosphatase and mineralization. Furthermore, the bone-healing properties were also improved in a fracture healing model according to the results of micro-CT, mechanical tests, and histological analysis [77]. Another working group analyzed plasma miRNA levels at days 3 and 14 after injury in a rat femoral bone defect model. They found that bone injury led to a significant upregulation of let-7a, let-7d, and miR-21-5p in plasma and splenocytes at day 14 relative to day 3 after bone injury, but not in sham-operated animals [78]. The positive effects of miR-21 on osteogenesis are also underlined by results from Oka et al. Bone samples from miR-21-knock-out mice had significantly lower levels of ALP and RUNX2, supporting the previously mentioned studies in the in vitro section [79].

### 5.3. Patient Studies

Regarding the miRNA analysis in patients with bone injury, the role of miR-21-5p was examined the most. In patients with vertebral fractures, the levels of miR-21-5p were significantly upregulated in older women with osteoporotic vertebral fractures, and/or low bone density (BMD) (regardless of osteoporosis treatment) [80]. Furthermore, miR-21/miR-21-5p was upregulated in the serum, as well as in the bone tissue [81,82,83].

Panach et al. suggested serum miR-21-5p as a valuable biomarker for the increased risk of fractures in patients [84]. This is also underlined by Kelch et al., who analyzed the miRNA expression in bone tissue, osteoblasts, and osteoclasts of 28 osteoporotic patients and found that miR-21-5p was increased in osteoporosis patients in osteoblasts and osteoclasts. Furthermore, a correlation of serum levels of miR-21-5p with BMD (bone mineral density) was found in this study [85]. In contrast, miRNA analysis from serum samples of postmenopausal women with low bone mass and vertebral fractures showed that miR-21, miR-23, and miR-29 were significantly downregulated compared to controls. The expression of miR-21-5p was significantly lower in osteoporotic/osteopenia women with vertebral fractures, showing 66% sensitivity and 77% specificity in distinguishing women with a vertebral fracture [86]. MiR-21-5p was downregulated in blood samples from osteoporosis patients compared to healthy patients with acute hip fractures [87].

In summary, in some studies, elevated miR-21 levels were associated with osteoporosis and a higher risk of developing fractures. Nevertheless, there are other studies highlighting the association between miR-21 downregulation and BMD, the development of osteoporotic vertebrae, and hip fractures. On the other hand, some studies show that miR-21 mostly seems to have a possible effect on osteogenesis and fracture healing. In particular, miR-21-5p has effects on osteogenesis with its influence on osteogenic pathways like the upregulation of bone morphogenetic proteins. Unfortunately, analyses comparing miR-21-5p and miR-21-3p are not available. A future study could aim to explore the impact of miR-21-3p on these proteins as well as differentiate the impact of miR-21 on Smad 7. Looking at osteoporotic fractures, findings seem consistent that miR-21 is upregulated, but it remains unclear what the exact impact on miR-21-3p is. It has to be taken into consideration that there could be differences between acute trauma and chronic conditions like osteoporosis. It is still unanswered if miR-21 improves bone healing or is upregulated due to the injury mechanism. To gain further clarity, more detailed studies are necessary, especially focusing on comparing the two variants in the same samples and conditions.

While most studies have focused on miR-21-5p, few studies justify their analysis of the miR-21 type, and studies on miR-21-3p in bone injuries are still missing (Table 4).

## 6. MiR-21 in Cardiac Damage

Chest trauma and cardiac damage are two predictors of poor outcomes in polytraumatized patients [88,89]. Approximately 30,000 patients with blunt cardiac trauma are recorded each year in the US [90]. In total, 43% of trauma patients showed increased levels of troponin at admission to the emergency room, which correlated significantly with the mortality in these patients [91]. Additionally, 78% of patients with impaired left ventricular function after blunt chest trauma died [92]. Therefore, cardiac damage after trauma should be diagnosed and treated as fast as possible to avoid such a tragic course.

### 6.1. In Vitro Studies

Tang et al. analyzed the effects of Prostaglandin-E1 (PGE1) treatment on hypoxia-induced cardiac damage in vitro. Rat H9C2 myoblast cells and isolated primary cardiomyocytes were subjected to hypoxia for 6 h, reoxygenated for 6, 12, or 24 h, and then treated with PGE1. They showed that PGE1 upregulated miR-21-5p expression in rat cardiomyocytes and significantly diminished cell cytotoxicity and apoptosis and decreased the expression of IL-2, IL-6, P-p65, TNF-α, and cleaved caspase-3 [93].

The treatment of hypoxia–reoxygenation (HR)-treated H9c2 cardiomyocytes with salidroside significantly increased cell viability and decreased LDH release, whereas inhibition of miR-21 reversed these effects. Salidroside mitigated HR-induced oxidative stress as illustrated by the downregulation of ROS generation and MDA (malondialdehyde) levels and increased the activities of the antioxidant enzymes SOD (superoxidismutase) and GSH-Px (glutathione), all of which were abrogated in cells transfected with the miR-21 inhibitor [94]. When rat cardiomyocytes were cultured under hypoxia and ischemia conditions and treated with ginsenoside, downregulation of miR-21 and mirR-320 due to ginsenoside was observed [95].

In an in vitro model of septic cardiomyopathy, Xue et al. observed that LPS treatment of H9c2 cells reduced cell viability and downregulated miR-21-5p expression. miR-21-5p overexpression inhibited LPS-induced inflammatory damage to the H9c2 cells, and PDCD4 acted as a downstream target gene of miR-21-5p [96].

### 6.2. In Vivo Studies

To evaluate the role of miR-21 in acute cardiac infarction, a mouse model of cardiac infarction with co-injection of agomiR-21 and agomiR-146a was performed. MiR-21 and miR-146a synergistically decreased apoptosis under ischemia/hypoxic conditions in cardiomyocytes. Co-injection of these two molecules led to a decreased infarct size and increased ejection fraction, as well as an increased shortening fraction. All of the effects were increased by double injection. The synergistic effect was mediated via miR-21-PTEN/AKT-p-p38-caspase-3 and miR-146a-TRAF6-p-p38-caspase-3 signal pathways [97].

Besides injury-induced cardiac damage, drug-induced damage to the heart is another important topic. In a drug-induced cardiac damage model in rats, treatment with either isoproterenol or allylamine upregulated miR-21-5p in cardiac tissue and localized it specifically to inflammatory cell infiltrates in the heart [98]. Moreover, a single sub-lethal dose of chest radiation (25 Gy) has been shown to result in compensatory upregulation of the myocardial connexin-43 (Cx43) and activation of the protein kinase C (PKC) signaling, along with a decline in miR-1 and an increase in miR-21 levels in the left ventricle (LV) [99].

In another mouse model of induced cardiac damage, cardiotoxicity was induced with doxorubicin. Exosomal miR-21 was then delivered to the heart with ultrasound-targeted microbubble destruction (UTMD) assistance. miR-21 delivery into the heart significantly decreased cell death, restored cardiac function, and partially reversed the doxorubicin-induced damage [100]. In a rat model of heat stress-induced heart illness, Sadat-Ebrahimi et al. identified multiple differentially expressed miRNAs in the heart, liver, kidney, or lung and showed in vitro that the inhibition of miR-21-3p in cultured cardiomyocytes increased the number of apoptotic cells five hours after heat stress [101].

### 6.3. Patient Studies

Our working group recently published that the systemic, and not the exosomal, concentration of miR-21-5p might be a useful tool for diagnosing cardiac damage in polytraumatized patients [102].

In another study with single-, dual-, and multivessel occluded coronary artery disease (CAD) patients, the expression of plasma miR-21 was evidently and progressively higher, while the programmed cell death protein 4 (PDCD4) level was significantly and steadily lower compared to healthy controls [103].

In the same study, human umbilical vein endothelial cells (HUVECs) were exposed to HR. A significant upregulation of miR-21 in HR-exposed HUVECs was observed, while the PDCD4 concentration was downregulated compared with control cells. The inhibition of miR-21 strongly reduced caspase-3 activity and ROS concentration while significantly ameliorating HUVEC cellular viability in HR conditions via enhanced PDCD4 expression [103]. Based on this study, miR-21 seems to negatively affect cell viability after hypoxia in endothelial cells.

It was suggested that miR-21 might play an important role not only in myocardial infarction or induced cardiac damage but also in acute heart failure (HF). This was supported by the findings of elevated plasma levels of miR-21 in patients with HF. However, in patients with ischemia-induced HF, the levels of miR-21 and miR-23 were significantly lower than in controls. The authors were not able to prove a significant association between these miRNAs and prognostic outcomes including in-hospital mortality, one-year mortality, and the number of readmissions [104].

Taken together, miR-21 seems to be upregulated in situations of cardiac damage and has mostly positive effects (Table 5). However, further studies are needed to evaluate the prognostic potential of miR-21 as a potential biomarker in these cardiotoxic entities.

## 7. Lung Injuries

Historically, acute respiratory distress syndrome (ARDS/ALI) was reported as one of the main causes of trauma-related mortality with rates up to 40%. With the improvement of critical care, the mortality of ARDS has decreased significantly in recent years. ARDS-related deaths were described in 6-20% of polytraumatized patients [105,106,107].

### 7.1. In Vitro Studies

No study was found to be focused on miR-21 in lung injury in vitro.

### 7.2. In Vivo Studies

MiR-21 seems to be upregulated in lung fibrosis, as shown in mice with bleomycin-induced lung fibrosis and in patients with idiopathic pulmonary fibrosis. Furthermore, miR-21 increases the pro-fibrogenic activity of TGF-ß in fibroblasts [108] and negatively regulates the NFκB inflammatory response in acute lung injury in rats [109].

### 7.3. Patient Studies

In radiation-induced lung injury, miR-21 increases the expression of TGF-ß1, which excludes some therapeutic options [110]. In patients with acute respiratory distress syndrome, the serum concentration of miR-21 was elevated [111]. Liu et al. compared miR-21-3p serum levels in patients with different severities of acute lung injury and found a positive link to mortality and the APACHE II score [112].

In summary, detailed information about the function of miR-21 is not available in lung injury. There are some hints pointing to an upregulation of miR-21 in injured lungs and a possible activation of apoptosis-related pathways (Table 6).

## 8. miRNA-21 in Polytrauma

Polytrauma is the leading cause of death and disability worldwide. Each year, over five million people across the globe pass away due to injuries resulting from traffic accidents, falls, drowning, burns, violence, and acts of war. Although the mortality of road traffic accidents has decreased constantly over the last few years, the mortality is still high. TBI was indicated as the most prevalent cause of death in polytrauma patients [113,114,115,116]. Other noteworthy facts about polytrauma patients are long-lasting impairments and declines in the quality of life [117].

In general, polytraumatized patients are defined as having either a life-threatening combination of injuries or one injury alone that is declared life-threatening. Another definition of polytraumatized patients involves an injury severity score equal to or above 16, based on the three most injured body regions, rated by a number in a square. Hemorrhage, thoracic injuries, and TBI are major causes of death in these patients. Later in the trauma response, multiple organ failure (MOF) or acute lung failure are predominant and life-threatening [118,119].

One thing all the polytrauma definitions have in common is the fact that the severity of a polytrauma is not only defined by the sum of the single injuries. The immunopathology is generally more complex than expected at first sight. On molecular levels, many different pathways are activated after severe trauma, leading to immune dysfunction and infectious complications [119,120]. The immunopathology of polytrauma is characterized by a pro- and anti-inflammatory response of the immune system at the same time, which causes complex regulatory processes in severely injured patients. Not surprisingly, miR-21 is also part of this complex molecular picture after trauma.

For example, miR-21 levels are correlated with the appearance of adverse events in recovery after multiple traumas and might therefore be a predictor of negative outcomes in severely injured patients [121]. Only a small number of existing studies have dealt with the role of miR-21 in the complex immune dysfunction after polytrauma. These suggest that the role of miR-21 is neither simply pro- or anti-inflammatory, nor obviously good or bad for patients—as also demonstrated in monotrauma situations.

### Patient Studies

As previously described, polytrauma is marked by immune dysfunction with an increased probability of infections in the course of treatment. A feared complication in the context of polytrauma is the development of sepsis. Xu et al. found that in severely burnt patients, miR-21 expression was reduced and seemed to be related to the degree of inflammation [122]. Recently, Weber et al. published data showing a decrease in miR-21-5p in exosomes that was associated with the development of sepsis (*p*
≤ 0.05) [123]. This suggests that miR-21 is an important player in the pathophysiology of septic complications after trauma. Furthermore, miR-21 might be a future tool for detecting cardiovascular complications by detecting cardiac damage in polytraumatized patients [102].

The expression profile of miRNAs seems to not only play a role in the acute trauma phase but also during the recovery and healing phases. Thus, miR-21-3p was positively correlated with a complicated recovery after polytrauma. Higher levels of miR-21 were observed at the time point of admission in patients suffering from ventilator-associated pneumonia during recovery [121].

Wang et al. identified miRNA-21 as a regulator in wound healing, and found that antagonizing miR-21 delays wound healing and impairs collagen deposition [124]. The TBI section of the present literature overview can hint at the conclusion that miR-21 could have positive effects on the outcome of severely injured patients. Unfortunately, insights on the role of miR-21 in the pathophysiology of polytrauma—especially the combination of TBI and other injuries—are rare, and further studies are needed to speak to this assumption. This is critical because TBI was reported to be the most prevalent cause of death in polytrauma patients [125].

To sum up, there are few studies regarding miR-21 in polytrauma patients. High levels of miR-21 seem to be associated with adverse events in the recovery after polytrauma. Higher levels of miR-21-5p in exosomes were associated with the development of sepsis. In contrast, higher levels of miR-21 were associated with better neurological outcomes in polytrauma patients with TBI (Table 7).

However, correlation analyses with data on serum levels of miRNA-21 in polytraumatized patients at different time points and with different complications do not exist. Due to the complex immune dysfunction in polytraumatized patients, the role of individual miRNAs remains unclear and challenging to decode. Research in this field promises to guide the future: miRNAs are potential biomarkers for various diseases, e.g., cancer, and could possibly be used as early markers for sepsis and the prediction of outcomes of patients.

## 9. Conclusions

The present review highlighted the various and ambivalent roles of miR-21 in acute organ injury (Figure 2). Concerning TBI, an increase in miR-21 is associated with better neurological outcomes, reduced apoptosis, and alleviated damage to the endothelial barrier or tight junctions. Moreover, miR-21 is associated with neuronal survival and better function recovery in the case of spinal cord injury. The development of neuropathic pain is associated with increased miR-21, while miR-21 shows anti-apoptotic effects in peripheral nerve injury. With regard to osteoporotic fractures, some authors described increased miR-21 concentrations while other authors observed a reduction in miR-21. In cardiac damage, the role of miR-21 is not clearly examined in different diseases. Serum levels of miR-21 were increased in patients with coronary heart disease and in patients diagnosed with heart failure. Increased research is necessary since the pathophysiological mechanisms and impact of miR-21 in cardiac disorders are not fully known. Concerning lung injury, miR-21 seems to be upregulated in lung fibrosis and able to inhibit apoptosis.

A largely ignored area is the possible therapeutic application of miR-21. Since miR-21 was shown to inhibit apoptosis in many diseases, it could be used as a therapeutic tool to decrease apoptosis and therefore preserve tissue. Another possible therapeutic application is functional recovery and neuronal survival in SCI patients induced by drugs elevating miR-21. In the case of polytrauma patients, who have different combinations of injuries, the use of miR-21 as a therapeutic target could be very complicated and will require individualized assessments. As this review has shown, miRNA-21 may be advantageous for certain types of injuries and detrimental for others. For instance, patients who have experienced polytrauma with both a pulmonary and a spinal cord injury might require elevated levels of miRNA-21 for the spinal cord damage and decreased levels for the lung injury at the same time. In summary, miRNA-21 has positive and negative effects in different acute organ injuries and increases or decreases in various diseases. This makes miRNA-21 currently challenging and non-advisable as a biomarker, but interesting for mechanistic analysis in the field of trauma research.

## Figures and Tables

**Figure 1 ijms-25-11282-f001:**
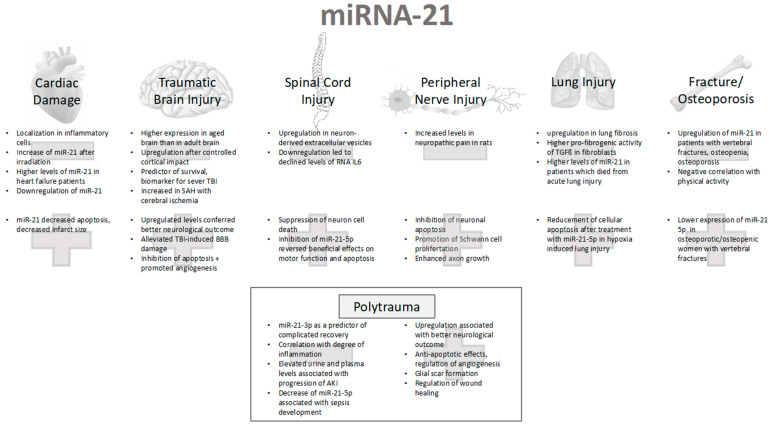
Overview of miRNA in different organ systems.

**Figure 2 ijms-25-11282-f002:**
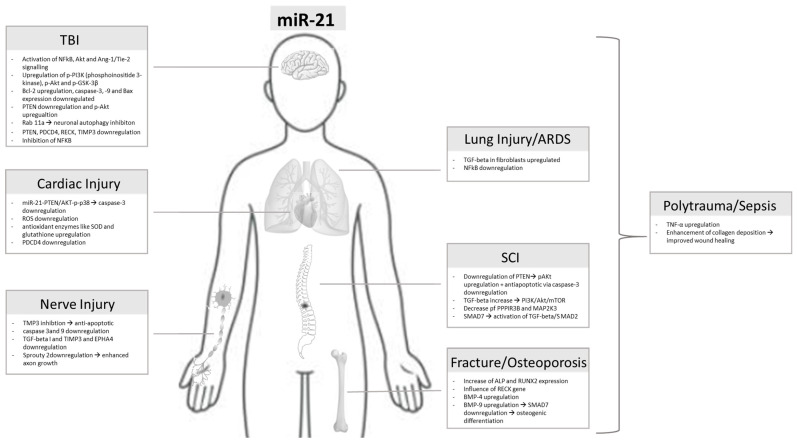
Overview of signaling pathways of miR-21 in organ injuries.

**Table 1 ijms-25-11282-t001:** Traumatic brain injury.

Scratch injury model on cultured brain microvascular endothelial cells (BMVECs), PC12 cells	Upregulation of miR-21-5p levels in BMVECs alleviated endothelial barrier damage and loss of tight junction proteins. miR-21-5p alleviated leakage of injured brain microvascular endothelial barrier. Suppression of inflammation and apoptosis through miR-21-5p. Impact on NF-kB, Akt, and Ang-1/Tie-2 signaling.	[16,17,18]
Scratch injury model on cultured neurons Transfection of miR-21 agomir/antagomir	Reduction in TUNEL-positive neurons after upregulation of miR-21. miR-21 decreased expression of PTEN. After transfection of miR-21 agomir in neurons, Bcl-2, caspase-3 and -9, and Bax expression promoted.	[19]
Coculture of PC12 (neuron) and BV2 (microglia) cells	Microglia polarization was induced by PC12-derived exosomes containing miR-21-5p. Polarization of M1 microglia led to release of neuroinflammation factors, inhibited neurite outgrowth, increased accumulation of P-tau, and promoted apoptosis of PC12 cells.	[20]
Cortical primary neuronal and astrocytic cells in oxygen- and glucose-deprived (OGD) medium	Upregulation (2-fold) of miR-21 in hypoxia	[21]
Fluid percussion injury rat model	Upregulation of miR-21 was linked to the following: Better neurological outcome. Inhibition of apoptosis and promotion of angiogenesis. Improvement of long-term neurological function. Alleviation of brain edema.	[17,32]
TBI mimic model: cultured HT-22 neurons treated with mouse TBI brain extract	Increased expression of miR-21-5p in isolated exosomes. miR-21-5p-enriched exosomes attenuated trauma-induced nerve injury through targeting Rab 11a.	[22]
Controlled cortical impact injury in adult and aged mice, miRNA analysis of injured cortex	Basal miR-21 expression higher in aged brain. In adult brain, miR-21 expression increased after injury, maximum increase 24 h after injury, returning to baseline 7 days post-injury. In aged mice, miR-21 showed no injury response, expression of miR-21 target genes (PTEN, PDCD4, RECK, TIMP3) was upregulated at all post-injury time points, maximal increase at 24 h post-injury.	[24]
TBI models in rats and mice	Upregulation of miR-21 expression at various time points after trauma. Elevation of miR-21 sustained 1 day to 7 days after trauma. Upregulation of miR-21 in hippocampus tissue after TBI.	[23,25,26,27,28,29,30]
Rat model of subarachnoid hemorrhage (SAH), transfer of MSC-derived EVs	MSC-EVs ameliorated early brain injury after SAH by reducing the apoptosis of neurons. Increase in miR-21 in SAH in prefrontal cortex and hippocampus.	[31]
Brain microvascular endothelial cells (BMVECs), cortical impact on mouse brain	Suppression of cell death after miR-21-3p antagomir transfection by inhibiting activity of NF-κB signaling. Intracerebroventricular infusion of miR-21-3p antagomirs promoted the expression of tight junction proteins, contributing to improved neurological outcome.	[33]
TBI patients	miR-21 as biomarker for the diagnosis of severe TBI. Prediction of outcome in TBI patients.	[35,36]

**Table 4 ijms-25-11282-t004:** Bone injuries.

Rat bone marrow-derived mesenchymal stem cells (rBMSCs) and repair capacity in a rat closed femur fracture model with internal fixation	Upregulation of miR-21 led to increased expression of osteopontin and alkaline phosphatase in rBMSCs, as well as promotion of mineralization.	[77]
Levels of circulating miRNAs, analysis of bone tissue samples in patients with fractures	miR-21-5p was significantly upregulated in patients with vertebral fractures. Upregulation of miR-21 in serum of patients with osteoporosis and osteopenia. Higher expression of miR-21 in bone tissue of osteoporotic patients. Negative correlation of miR-21 levels and moderate physical activity.	[81,82,83]
Analysis of miRNA from serum samples, bone tissue analysis	miR-21-5p upregulated biomarkers in patients with increased risk of bone fracture. miR-21-5p was significantly upregulated in patients with vertebral fractures. miR-21-5p was increased in osteoporotic patients in osteoblasts and osteoclasts. Positive correlation of miR-21-5p serum levels and BMD.	[80,84,85,86]

**Table 5 ijms-25-11282-t005:** Cardiac damage.

Rat H9C2 cells and isolated primary cardiomyocytes were cultured under hypoxic conditions and treated with Prostaglandin-E1 (PGE1)	PGE1 upregulated miR-21-5p expression.	[93]
Serum samples of polytrauma patients with and without increased troponin levels	Exosomal concentration of miR-21 was increased in polytrauma patients with increased troponin concentration at the emergency room time point.	[102]
Patients with coronary artery disease (CAD) In vitro hypoxia–reoxygenation (HR)-exposed HUVECs	Plasma levels of miR-21 were increased in CAD patients. PDCD4 levels were decreased in CAD patients. Expressions of miR-21 in HR-exposed HUVECs were upregulated. PDCD4 concentrations were downregulated in the same cells.	[103]
Mouse model of cardiac infarction with co-injection of agomiR-21 and agomiR-146a	miR-21 decreased apoptosis under hypoxic conditions in cardiomyocytes. Co-injection decreased infarct size, increased ejection fraction (EF), and induced fractional shortening (FS). Synergistic effects via miR-21-PTEN/AKT-p-p38-caspase-3 and miR-146a-TRAF6-p-p38-caspase-3 signal pathways.	[97]
Plasma samples from patients with acute heart failure (HF) compared to healthy controls	Higher plasma levels of miR-21 in patients with heart failure. No association of miRNA levels and outcome parameters.	[104]

**Table 6 ijms-25-11282-t006:** Lung injury.

Acute lung injury in rats	Negative regulation of the inflammatory response by miR-21.	[109]
miRNA serum levels in patients	Increased serum levels of miR-21-3p in patients who later died and patients with higher Apache score. Increased miR-21 concentration in acute respiratory distress syndrome patients.	[111,112]

**Table 7 ijms-25-11282-t007:** Sepsis/trauma.

miRNA analysis in plasma samples of polytraumatized patients	miR-21-3p as a predictor of complicated recovery. Decrease in miR-21-5p in exosomes was associated with development of sepsis.	[121,123]
miRNA analysis in plasma samples of patients after burn with and without sepsis	Reduction in miR-21 expression in serum of patients with sepsis after burns.	[122]
Peritonitis in cecal ligation mouse model	Overexpression of miR-21 decreased TNF-α secretion. miR-21 seems to be beneficial for survival in sepsis.	[126]
Skin wounds were treated with miR-21 antagomir	Antagonizing of miR-21 led to delayed wound healing and impaired collagen deposition.	[124]

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
