# Peer review of "The Ambivalent Role of miRNA-21 in Trauma and Acute Organ Injury"

_ijms, 2024, doi:10.3390/ijms252011282_

Round 1

Reviewer 1 Report

Comments and Suggestions for Authors

Ritter et al submit a review entiteld "The ambivalent role of miRNA-21 in trauma and acute organ injury"

They focus on this really interesting miR and make a nice balance between its benefitial and pathological roles.

This reveiw is well written, compleate. It desperately needs figures; Tables are allright but a review needs to be illustrated.

two suggestions; 1. a translational figure on the differents roles of miR21 in acute organ damage after trauma on the various organs. 2. different ways to vectorize miR-21 such as nanotech, RNA modifications, crispR etc.

Reviewer 2 Report

Comments and Suggestions for Authors

The manuscript aims to provide a comprehensive review of the dual roles of miRNA-21 in trauma and acute organ injury, with a focus on traumatic brain injury, spinal cord injury, peripheral nerve injury, bone injuries, cardiac damage, and lung injuries. This review is rich in content and logically clear. However, there are still some details and analyses that need further improvement and deepening.

1. This manuscript lacks depth in mechanistic insights. Is miRNA-21 subjected to similar regulation in trauma or acute organ injury? Do the genes regulated by miRNA-21 have some common characteristics, including sequence and function?

2. While the review mentions the pathways and targets involved (e.g., PTEN/AKT/mTOR pathway, TGF-β signaling), it lacks a detailed mechanistic discussion of how miRNA-21 modulates these pathways in different contexts.

3. "The literature research was conducted from the 14.08.2023 until the 01.05.2024." Why did you choose literature between "14.08.2023" and "01.05.2024"? These two dates seem arbitrary or inconsistent. Additionally, "01.05.2024" should be written as "5 October 2024" to avoid confusion. I am a bit confused about the references used in the manuscript, as they are not limited to this specific time period.

4. miR-21 has two isoforms: miR-21-5p and miR-21-3p. Their different functions should be emphasized. In many parts of the manuscript, it is not clearly specified which isoform is acting. I hope the authors will carefully read the original papers and then indicate this.

5. The table numbering should be "Table 1," "Table 2," rather than "Table 1.1," "Table 1.2." The tables lack captions. Additionally, tables lack proper referencing in the text, and some tables are not sufficiently detailed or are redundant with the main text.

6. The authors need to review the manuscript for grammatical errors and improve the clarity of writing.

Comments on the Quality of English Language

The authors need to review the manuscript for grammatical errors and improve the clarity of writing.

Round 2

Reviewer 1 Report

Comments and Suggestions for Authors

Changes ok

Reviewer 2 Report

Comments and Suggestions for Authors

The authors have satisfactorily addressed my suggestions, and for the issues that could not be easily resolved, they have provided reasonable explanations in their response letter. However, I would like to encourage the authors to carefully review the manuscript once more to ensure that there are no grammatical errors or typos.